# CoLLEGe: Concept Embedding Generation for Large Language Models

**Ryan Teehan, Brenden M. Lake & Mengye Ren**
New York University
{rst306,brenden,mengye}@nyu.edu

## Abstract

Current language models are unable to quickly learn new concepts on the fly, often requiring a more involved finetuning process to learn robustly. Prompting in-context is not robust to context distractions, and often fails to confer much information about the new concepts. Classic methods for few-shot word learning in NLP, relying on global word vectors, are less applicable to large language models. In this paper, we introduce a novel approach named **CoLLEGe** (**Co**ncept **L**earning with **L**anguage **E**mbedding **Ge**neration) to modernize few-shot concept learning. CoLLEGe is a meta-learning framework capable of generating flexible embeddings for new concepts using a small number of example sentences or definitions. Our primary meta-learning objective is simply to facilitate a language model to make next word predictions in forthcoming sentences, making it compatible with language model pretraining. We design a series of tasks to test new concept learning in challenging real-world scenarios, including new word acquisition, definition inference, and verbal reasoning, and demonstrate that our method succeeds in each setting *without task-specific training*. Code and data for our project can be found at https://college-concept-learning.github.io/.

## 1 Introduction

Imagine a student first attending a philosophy lecture on epistemology, wherein their professor discusses and critiques the positions of idealists, pragmatists, and foundationalists, among others. Some concepts and terms, such as idealism or pragmatism, may be familiar from past experience but in this new and unfamiliar context they seem to have taken on new meaning. Other concepts may be entirely new, including the concept of "epistemology" itself. During the lecture, the examples the professor provides for each concept, as well as the sentences they use when discussing them, allow the student to form an initial sense of their meaning and usage. With time, additional examples, and using the concepts directly in writing, the student's knowledge solidifies and what was once unfamiliar is now intuitive.

Building intuitions about the meaning of unfamiliar concepts in this way, with only a few examples of their usage, is common in real-world human learning, but remains difficult for language models, particularly when we want to consolidate this knowledge into discrete tokens. Providing a few in-context examples of how to use these new tokens can be a stopgap, but is often less powerful the additional examples can serve as distractors that confuse the language model, to say nothing of how unnatural it is (imagine if, each time the professor wanted the student to answer a question about epistemological idealism, they began by repeating a few sentences containing "idealism"). Instead, with a few example sentences, language models should know general semantic knowledge about this new concept, similar to the knowledge encoded in their pretrained representations. We frame this as a few-shot learning problem, where, given a few example sentences, the goal is generate an embedding for a new concept token with *expressive and task-general* semantic information.

Prior work on few-shot word learning in natural language focused on leveraging the seminal works on global word vector representations (Mikolov et al., 2013; Pennington et al., 2014;

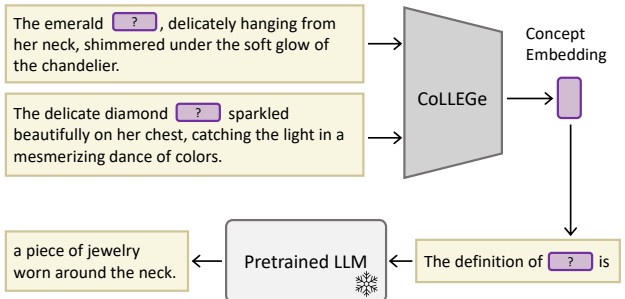

Figure 1: Our model generates an embedding for an unseen token given one or a few example sentences. The ground truth word is *pendant*, and the model is able to generate an accurate definition using the embedding produced by CoLLEGe.

Khodak et al., 2018; Lampinen & McClelland, 2017). However, these methods are less well suited to augmenting the knowledge of contemporary large language models. First, the embeddings generated from methods based on global word vector representations may be difficult to adapt to the representation space of contemporary language models. Additionally, learned contextual representations from pretrained language models provide a more powerful and semantically rich source for few-shot concept learning, allowing for complex usage of new concepts using embeddings generated from only a few examples.

Furthermore, prior evaluation methods for new concept learning were limited to noisy proxy measures, such as the correlation between embedding cosine similarity and human similarity judgements (Lazaridou et al., 2017), or the cosine similarity between the few-shot concept embeddings and "ground truth" Word2Vec embeddings (e.g. the definitional nonce in Herbelot & Baroni (2017)). Current language models are able to use language in highly sophisticated and complex ways; true evaluation of few-shot concept learning in this setting should assess whether new concepts can be used in similarly complex and sophisticated ways. What older evaluations measure tell us little about how well language models can use learned concepts. How well can they define a concept given only a few examples? Can they correctly answer fill-in-the-blank questions for difficult words? We ask humans the same questions to determine how well they have internalized a new concept. Our evaluations of language models should follow suit.

In this paper, we present CoLLEGe, a novel and conceptually simple framework to enable large language models to quickly learn new concept tokens. To evaluate our method, we develop a suite of tasks to directly evaluate how well these concepts are learned, including GRE-style fill-in-the-blank verbal reasoning, definition inference and generation, and Internet slang usage. Our method leverages the vast amount of pre-training data and learning can be seamlessly embedded in the model pre-training process. We discover that training techniques such as an example buffer, negative example sampling, and knowledge distillation contributed significantly to the model's concept learning performance. Moreover, thanks to our general pre-training procedure, our model is able to transfer to these concept learning tasks in a zero-shot manner with no additional task-specific training needed, while maintaining the LLM's original performance on regular data.

In summary, our contributions are:

- A simple add-on learnable module for few-shot, LLM concept learning;

- An approach to training our algorithm which combines an example buffer, negative sampling, and knowledge distillation. We show that each of these components plays an important role in learning;

- Three challenging datasets – CoLLEGe-GRE, CoLLEGe-DefGen, and CoLLEGe-Slang – used to measure the effectiveness of few-shot concept learning methods for LLMs. These datasets test both general and complex concept knowledge, naturalistic acquisition of new concepts, and relational abstraction;

- Experiments showing that, by training an embedding generation modules in a task-general manner, we can generate embeddings that allow a pretrained LLM to: **a)** generate plausible definitions for new concepts, **b)** correctly solve fill-in-the-blank tasks for difficult

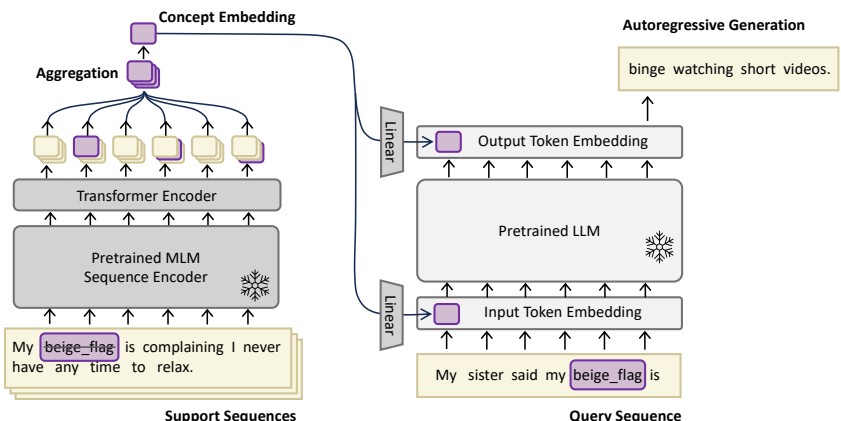

Figure 2: Our proposed CoLLEGe framework for concept embedding generation. Support sequences are embedded by a pretrained MLM (e.g. RoBERTa) with an additional Transformer encoder to produce pooled sequence embeddings for each support sequence. These are aggregated and projected into the input and output embedding space for the pretrained LLM (e.g. LLaMA). According to UrbanDictionary, *beige flag*, an Internet slang appeared in mid-2023, means "a benign but annoying trait or habit."[2]

words, and **c)** correctly identify the meaning of new slang terms without additional training.

## 2 Related Work

**Few-Shot Word Learning:** A classic and related task in NLP is rare word learning (Luong et al., 2015). Lazaridou et al. (2017) create the synthetic "chimera" concepts and provide early evidence that summation over (global) word vectors in the context surrounding a new or rare word can produce a useful embedding. Khodak et al. (2018) build on this, presenting a method which includes a learned linear transformation to account for shared features across global word vectors. Lampinen & McClelland (2017) present an even simpler method, involving freezing the majority of the weights of the network and using gradient descent to tune only the weights related to the new word(s). For a more complex approach, Herbelot & Baroni (2017) modify the Word2Vec algorithm for more effective few-shot learning. More modern approaches include HiCE (Hu et al., 2019), which uses Transformer layers to induce a new word embedding from Word2Vec embeddings, and Mem2Vec (Sun et al., 2018), which uses a long-term memory system. Similarly, Weston et al. (2015) model new words by using contextual examples from memory. They store a bag-of-words for the left and right context surrounding the new word, and simulate new word learning with a fixed percent of words encountered during training. Instead of using global word vectors to represent the context, we use the contextual representations from a frozen MLM to learn a representation for the new concept token. We also combine autoregressive losses, cosine distance, and distillation loss to train our model, whereas older work either reused the Skip-gram objective or used cosine distance. Other approaches incorporate morphological information (Luong et al., 2013; Schick & Schütze, 2019). While these methods are useful, particlarly for learning global word vector representations, they are less useful for augmenting the embeddings of pretrained LLMs. In part, this is because global word vectors do not map easily to the pretrained LLM embedding space. Additionally, global word vector representations are often less informative than the pretrained representations from BERT-style models.

**Meta-Learning:** Matching Networks (Vinyals et al., 2016) and Prototypical Networks (Snell et al., 2017) both approach few-shot learning as a meta-learning problem. Online prototypical networks (Ren et al., 2021) build on the latter for novel concept learning in an online and continual fashion. Our approach is also related to fast-weight networks (Schmidhuber, 1992), since we use the support sequences to generate a fast embedding weight for the new concept. For language, meta-learning is often used for knowledge augmentation (Hu et al., 2023), task adaptation (Chen et al., 2022; Zhong et al., 2021; Bansal et al., 2020), domain adaptation

---

[2]https://www.urbandictionary.com/define.php?term=Beige%20flag

(Qian & Yu, 2019; Li et al., 2020; Geng et al., 2019), rare word recognition (Lux & Vu, 2021) (in ASR), and word sense disambiguation Holla et al. (2020), among other applications (Lee et al., 2022). Some recent methods frame meta-learning as a sequence modeling problem, drawing inspiration from in-context learning (Chen et al., 2022; Fifty et al., 2024). Finally, Lake & Baroni (2023) recently developed a method for learning compositional concepts. In our work, context sentences are encoded to generate a new embedding, conceptually similar to a prototype for the new concept, to optimize a general language modeling objective, rather than a collection of task objectives.

**Compression:** A number of methods exist to compress sentences into new embeddings or tokens. Prior work in NLP developed methods for generating task-general embeddings from natural language sentences (Conneau et al., 2017; Kiros et al., 2015; Wang et al., 2020). ReadOnce (Lin et al., 2021) is a more recent method for generating compressed document representations which can be used across a variety of downstream tasks. Similarly, recent methods compress prompts (Chevalier et al., 2023; Ge et al., 2024; Mu et al., 2024) or documents Xu et al. (2024) into summary vectors either as a form of memory or a method for re-using prompts (e.g. when specifying instructions). RMT (Bulatov et al., 2022) learns memory tokens during pretraining in order to extend the effective context window. Nugget (Qin & Van Durme, 2023) dynamically chooses which tokens are aggregated into the encoded representation. Rather than compressing the entire meaning of each context sentence, our method extracts and aggregates information relevant to the new concept.

## 3 CoLLEGe: Concept Learning with Language Embedding Generation

In this section, we describe our proposed approach for enabling LLMs to quickly learn new concept tokens. Given a new word or concept and a set of example sentences containing that word or concept, we want to produce an embedding that captures its semantically meaningful features. We visualize this process for a simple language modeling example in Figure 2.

Framing this as a few-shot learning problem, we use a set of *K support sequences* $\{s_1, ..., s_K\}$, containing a new token, <nonce> to produce a useful embedding for this new token. The new embedding can then be used to augment the knowledge of a frozen autoregressive language model. During training, we encourage the LM to use the new embedding to correctly generate a query sequence $q$.

**Concept Embedding Generation:** To perform this generation, the new token in each support sequence is replaced with a <mask> token, and each is embedded with a frozen masked language model (MLM) used for feature extraction. The contextual embeddings for each sequence are then passed through an additional Transformer self-attention layer to process the contextual embeddings for each sequence to obtain $\{h_{i,t}\}$. These are then aggregated using mean pooling, producing $k$ sequence embeddings $\{e_1, ..., e_k\}$: $e_i = \frac{1}{n_i} \sum_{t=1}^{n_i} h_{i,t}$, where $n_i$ is the length of each sequence. The sequence embeddings are aggregated once more using mean pooling, producing a single output embedding $e_{\text{new}}$: $e_{\text{new}} = \frac{1}{K} \sum_{i=1}^{K} e_i$. Mean pooling can also facilitate incremental consolidation of new concepts without having to store past examples. To integrate the embedding with a frozen autoregressive LM, we apply two distinct linear layers to produce an input and output embedding for the new token $e_{\text{in}}$ and $e_{\text{out}}$: $[e_{\text{in}}, e_{\text{out}}] = \text{Linear}(e_{\text{new}})$. The autoregressive LM's input and output token embedding matrices are then expanded with these generated embeddings, and used to model the query sequence.

$$W_{\text{emb\_in, new}} = [W_{\text{emb\_in}}, e_{\text{in}}], \tag{1}$$

$$W_{\text{emb\_out, new}} = [W_{\text{emb\_out}}, e_{\text{out}}]. \tag{2}$$

**Sampling Few-Shot Learning Episodes:** One novel aspect of our framework is that, unlike many meta-learning approaches, our training procedure follows the same style as pretraining by directly leveraging text data from the pretraining datasets. We hypothesize that a good way to rapidly learn new concept is to actually "use" the concept in another sentence—we let an LLM consume the newly learned embedding to generate another sentence. Moreover, the autoregressive cross entropy loss is the same as the pretraining

objective, so, in theory, our meta-learning procedure can be perfectly blended into the pretraining stage.

To efficiently sample support and query sequences, terms we borrow from the few-shot learning literature, we save sentences containing a new token in an example buffer to serve as support sequences, and when we encounter the same token being used again in the training corpus, we will use the sequence as a query sequence. The query sequence can then be saved in the example buffer and used as a support sequence again. We find that reusing query sequences as support sequences is helpful for training and allows the model to make use of examples it has already learning when learning new examples. Often, query sequences are longer, comprising a few sentences or a paragraph of text. The sentence which contains the new token is extracted from each and used as a support sequence for a different query sequence concerning the same concept. Not every such sequence ends up in the final set, however, since we filter and rank the examples, see Section 4 for details.

**Negative Examples:** Initial experiments training on only positive examples, i.e. examples containing the new token, tended to yield generated embeddings with norms of much higher magnitude than those of other LLM input or output token embeddings. One hypothesis was that, since every example the model has to learn contains a new token, it prefers to converge an embedding with high norm, to ensure that the new token appears in the query sequence. During normal pretraining, few tokens are shared across all sequences, and language models learn both when to generate and when not to generate each token. To likewise teach our model when not to generate a new token, we sample a sequence without a new token, which we call a **negative example**, and take the sum of the cross entropy loss on the positive example, $L_{\text{ce}}^{+}$, and the cross entropy loss on the negative example, $L_{\text{ce}}^{-}$.

**Knowledge Distillation:** Since we pretrain by replacing existing words with new tokens, we can compute the "true" language model embeddings and logits for the remainder of the sequence. Ideally, we want the generated embeddings from our model to match the ground truth embeddings and logits as faithfully as possible, to better approximate the underlying language model distribution. To do this, we retain the original sequence, before masking a word with a new token, and compute the output embeddings and logits for the rest of the sequence with a non-augmented version of our pretrained autoregressive model. We then compute the cosine distance between those output embeddings and the output embeddings from CoLLEGe, $L_{\text{cos}}$, as well as the MSE between the CoLLEGe LLM logits and the "true" LLM logits using the original embeddings, $L_{\text{mse}}$. Using a more standard objective with a distillation temperature (Hinton et al., 2015) was less effective during training.

In order to compute these two distillation loss terms, we define the positive example token sequence as $t_{1,+}, ...., t_{n,+}$ and original example token sequence as $t_{1,\text{orig}}, ...., t_{l,\text{orig}}$, and additionally construct a deterministic mapping $\sigma : \mathbb{N} \to \mathbb{N}$ that maps a token at index $i$ in the positive example to its corresponding token at index $k$ in the original sequence. The tokens following the new token in the positive sequence are guaranteed to have a match in the original example sequence, by definition, but the index may be different (if, for example, the word replaced with <nonce> is subtokenized in the original example sequence). We compute our distillation loss terms using:

$$L_{\text{cos}} = \frac{1}{n - |I_{\text{new}}|} \sum_{k \notin I_{\text{new}}} 1 - \cos(e_{t_{k,+}}, e_{t_{\sigma(k),\text{orig}}}), \tag{3}$$

$$L_{\text{mse}} = \frac{1}{n - |I_{\text{new}}|} \sum_{k \notin I_{\text{new}}} (\ell_{t_{k,+}} - \ell_{t_{\sigma(k),\text{orig}}})^2, \tag{4}$$

where $I_{\text{new}}$ is the set of new token indices in the positive example sequence, $E_{i,\cdot}$ denotes the output embedding at token position $i$ for the positive or negative example sequence, and $\ell_{i,\cdot}$ denotes the logit vector at token position $i$ for the positive or negative example sequence.

**Final Loss:** Our final training loss is simply a sum of these individual loss terms. We explored using a linear combination of the loss terms to weight each differently, but found

| Method | Without Definition | | | | With Definition | | | |
| | 1-Shot | 2-Shot | 3-Shot | 4-Shot | D+1-Shot | D+2-Shot | D+3-Shot | D+4-Shot |
|---|---|---|---|---|---|---|---|---|
| TT-1 | $6.8 \pm 0.0$ | $6.8 \pm 0.0$ | $6.8 \pm 0.0$ | $6.8 \pm 0.0$ | $10.9 \pm 3.0$ | $8.6 \pm 3.1$ | $8.1 \pm 3.9$ | $8.8 \pm 4.3$ |
| TT-2 | $6.8 \pm 0.0$ | $6.8 \pm 0.0$ | $6.8 \pm 0.0$ | $6.8 \pm 0.0$ | $12.3 \pm 3.5$ | $9.3 \pm 2.5$ | $8.1 \pm 5.3$ | $7.6 \pm 2.5$ |
| HiCE | $11.5 \pm 5.0$ | $12.5 \pm 2.7$ | $13.1 \pm 3.6$ | $16.1 \pm 4.2$ | $19.3 \pm 3.5$ | $15.9 \pm 8.2$ | $11.4 \pm 2.3$ | $10.6 \pm 1.3$ |
| Additive | $13.6 \pm 2.3$ | $9.1 \pm 3.9$ | $13.6 \pm 3.9$ | $11.4 \pm 0.0$ | $12.9 \pm 1.3$ | $10.6 \pm 6.6$ | $12.9 \pm 3.5$ | $7.6 \pm 1.3$ |
| Prompting | $13.9 \pm 4.3$ | $17.7 \pm 3.0$ | $19.8 \pm 4.2$ | $21.8 \pm 3.0$ | $21.6 \pm 5.7$ | $20.5 \pm 6.8$ | $19.3 \pm 4.8$ | $24.0 \pm 4.7$ |
| CoLLEGe w/o KD / Neg. Ex. | $32.2 \pm 5.2$ | $37.7 \pm 4.9$ | $38.6 \pm 2.9$ | $42.2 \pm 3.1$ | $40.0 \pm 6.6$ | $\mathbf{49.3} \pm 5.8$ | $\mathbf{49.6} \pm 3.5$ | $48.0 \pm 5.0$ |
| CoLLEGe w/o KD | $25.9 \pm 4.7$ | $33.6 \pm 5.0$ | $35.0 \pm 4.7$ | $35.7 \pm 3.2$ | $40.5 \pm 1.9$ | $40.2 \pm 3.8$ | $43.2 \pm 3.5$ | $42.3 \pm 3.2$ |
| CoLLEGe w/o $L_{\cos}$ | $30.9 \pm 6.0$ | $35.2 \pm 5.0$ | $36.4 \pm 4.2$ | $37.1 \pm 2.8$ | $35.7 \pm 2.5$ | $39.6 \pm 3.7$ | $39.3 \pm 5.2$ | $38.4 \pm 1.4$ |
| CoLLEGe w/o $L_{\mathrm{mse}}$ | $31.4 \pm 5.1$ | $38.2 \pm 4.6$ | $39.8 \pm 4.1$ | $43.6 \pm 4.6$ | $38.6 \pm 6.0$ | $39.6 \pm 4.5$ | $44.8 \pm 5.4$ | $46.8 \pm 3.4$ |
| CoLLEGe w/o Neg. Ex. | $28.9 \pm 6.0$ | $31.6 \pm 5.1$ | $34.1 \pm 4.8$ | $33.6 \pm 4.3$ | $37.5 \pm 6.2$ | $38.4 \pm 4.8$ | $42.5 \pm 4.6$ | $44.1 \pm 4.6$ |
| CoLLEGe | $\mathbf{35.0} \pm 5.6$ | $\mathbf{40.5} \pm 4.3$ | $\mathbf{42.7} \pm 3.9$ | $\mathbf{44.5} \pm 3.9$ | $\mathbf{42.5} \pm 3.9$ | $46.8 \pm 3.7$ | $45.2 \pm 3.2$ | $\mathbf{49.3} \pm 1.5$ |

Table 1: Accuracy in percentage on the GRE Verbal Reasoning task. For Token Tuning, we use LR = 1e-3, as that gave the best performance.

it was not significantly better. Our final loss, $L_{\text{total}}$, is:

$$L_{\text{total}} = \underbrace{L_{\text{ce}}^{+} + L_{\text{ce}}^{-}}_{\text{Cross Entropy Losses}} + \underbrace{L_{\cos} + L_{\text{mse}}}_{\text{Distillation Losses}} . \tag{5}$$

# 4   Datasets for Training CoLLEGe

In contrast to many other meta-learning methods, which use a specific set of tasks during training, we adopt a training approach that mirrors general-purpose pretraining. In a sense, we treat each query sequence, and each new token in turn, as its own "task" to solve. Pretrained language model representations are highly adaptable, and can be successfully applied to a variety of tasks with simple prompting strategies. By adopting a task-general training method, we train a module that can produce similarly adaptable embeddings on the fly.

Because CoLLEGe is designed to learn a single new token per sequence, and the LLM is frozen, training is highly sensitive to data quality, both for the support and query sequences. Additionally, three forms of mismatch between support and query sequences are important to guard against: language, contextual meaning, and knowledge mismatch. The first case is mostly self-explanatory, non-English support sequences for an English query sequence cause difficulties in training. Contextual meaning mismatch was particularly important to avoid when training with the Pile (Gao et al., 2020), whose examples are drawn from a variety of sources. Creating support and query sequences from WikiText, as in HiCE, often implicitly controls for contextual meaning (all examples are from one source (Wikipedia), and support and query sequences for a word are often unintentionally drawn from the same article and thus share contextual meaning). Likewise, knowledge mismatch is more prominent when training with the Pile, since it contains more diverse sources. If one, or many, support sequences are more confusing than the query sequence, this can destabilize training.

Using the Squeakily[3] library, we filtered for English text at threshold 0.90 using the FastText (Joulin et al., 2016) model for language identification, applied perplexity filtering at threshold 1000 (filtering examples above 1000 perplexity) using KenLM, following de la Rosa et al. (2022). We also filtered examples with too much character repetition as well as examples with words flagged as obscene. Afterwards, we cleaned examples by normalizing white space and punctuation. Each query sequence is constructed from 4 sentence, non-overlapping, chunks of the text examples from the Pile samples.

To build a set of support sequences for each query sequence, we first split all examples into individual sentences, and matched each query sequence with sentences that use the same new word. We removed sentences that appear in the query sequence, examples with a large number of newline characters (these often were article titles, tables of contents, or lists), and examples with fewer than 15 words. Earlier experiments training with the Pile showed that some subsets had many low-quality or mixed-meaning examples. Excluding those subsets made the filtering process more straightforward. Table 6 summarizes the top subsets from the Pile represented in our dataset.

# 5   Experiments

In this section, we show experimental results on four different evaluation tasks that we designed: GRE verbal reasoning, definition generation, and slang identification. Note that

---

[3]https://github.com/CarperAI/squeakily

| Model | BERTScore F1 | ROUGE–L | ELO | |
|-------|--------------|---------|-----|---|
| TT-1 | $75.2 \pm 8.1$ | $7.67 \pm 0.1$ | 980.78 | $\pm$ 18.5 |
| TT-2 | $75.2 \pm 8.1$ | $7.64 \pm 0.1$ | 978.49 | $\pm$ 18.4 |
| HiCE | $76.7 \pm 2.2$ | $7.98 \pm 0.0$ | 975.64 | $\pm$ 7.9 |
| Additive | $80.1 \pm 2.3$ | $11.66 \pm 0.1$ | 967.22 | $\pm$ 8.5 |
| Prompting | $82.5 \pm 2.8$ | $15.54 \pm 0.1$ | 1032.28 | $\pm$ 28.3 |
| CoLLEGe | $\mathbf{84.8} \pm 2.3$ | $\mathbf{17.81} \pm 0.1$ | **1065.57** | $\pm$ 24.0 |

Table 2: Results when evaluating definition generation using the CoLLEGe-DefGen dataset. We compare the model generated definitions with a reference definition using BERTScore and also compute an ELO score for head-to-head comparison. For Token Tuning, we use LR = 3e-4, as that performed best. Definitions are generated 1-, 2-, and 3-shot and scores are averaged.

CoLLEGe is a task-general concept embedding generation network, and all of the evaluations are performed *zero-shot*, *without* further training or fine-tuning, just like pretrained LLMs. In the following, we first discuss implementation and training details, then describe the baseline methods. Afterwards, we present the core results in Subsections 5.1-5.3.

**Implementation Details:** As our pretrained MLM model for the Sequence Encoder, we use RoBERTa-Large (Liu et al., 2019), and apply a trainable Transformer Encoder layer to encode the RoBERTa sequence embeddings. These embeddings are aggregated using mean-pooling to produce a single embedding per sequence, which is further mean-pooled into our Concept Embedding. We use a pretrained LLaMA-2 7B model (Touvron et al., 2023) as the pretrained autoregressive language model in all our experiments. Further implementation details can be found in Appendix B.

**Baselines:** In order to evaluate the effectivness of our method, evaluate against baselines from prior work on new concept learnign as well as prompting and gradient descent tuning. More details on implementation for the baselines can be found in Appendix E.

- **Token Tuning (TT)** (Lampinen & McClelland, 2017) finetunes only the new token embedding(s) using gradient descent. The support sequences are treated as the training batch for each step. TT-$N$ denotes $N$ gradient descent steps. Unlike Lampinen & McClelland (2017), $N$ is kept small here since we found that a large $N$ results in degraded performance. A similar approach has been proposed in Textual Inversion (Gal et al., 2023) for image few-shot learning and generation.

- **HiCE** (Hu et al., 2019) consists of a Transformer sequence encoder as well as a Transformer layer to aggregate the sequence embeddings. It is trained to output an embedding with minimal cosine distance to the true Word2Vec embedding.

- **Additive** (Lazaridou et al., 2017) is a simple baseline that consists of summing the Word2Vec embeddings for all tokens in the context that are not the new token.

- **Prompting** uses randomly initialized new token embeddings and includes the support sequences in the prompt. It is a strong baseline with direct context access. Since prompting allows the LM to reason over the tokens in context, it can be combined with our embedding generation approach (see Appendix F for additional experiments). However, prompting original sentences can also make the context window too long and distracting.

## 5.1 GRE Verbal Reasoning

The GRE verbal reasoning task is a challenging type of question appearing in Graduate Record Examinations that not only tests the understanding of rare vocabulary but also their logical placement in a sentence. We increase the difficulty here by making each multiple choice answer an unknown vocabulary word with a few example sentences as hints. We test whether the CoLLEGe generated concept embeddings can directly support downstream verbal reasoning.

**Dataset:** Using actual GRE practice questions from a Kaplan GRE prep book (Kaplan, 2019), we design a task where a language model has to either select the top or top-2 choices for a sequence with blanks. Examples for each of these questions are provided in Table 7 in Appendix C. Questions were hand-annotated from the Kaplan book and details about the cleaning process can be found in Appendix D. We produce a high-quality selection of 44

| Example Sentence | CoLLEGe Definition | True Definition | Word/ Phrase |
|---|---|---|---|
| The eerie creak of the attic door, coupled with the flickering candlelight, was enough to give anyone the <nonce>. | a feeling of unease, usually in the stomach, caused by anxiety or fear. | feelings of uneasiness | *willies* |
| Intrigued by holistic therapies, she found herself lying on a soft mat as the therapist applied <nonce> to various points on her body to alleviate her chronic migraines. | a substance that is used to heal or soothe a part of the body. | treatment of symptoms by applying pressure with the fingers to specific pressure points on the body | *acupressure* |
| Nestled in the far corner of the bustling newsroom, the diligent <nonce> worked tirelessly, transcribing reporter's notes into clean, easy-to-read articles. | a person who writes or edits for a newspaper, magazine, or other publication. | someone employed to make written copies of documents and manuscripts | *copyist* |
| The delicate <nonce> sprouted from the forest floor, adding a touch of alien beauty to the woodland scene. | a plant that resembles a mushroom. | a fungus composed of several apothecia that look like elongated rabbit ears; | *Wynnea americana* |

Table 3: Definitions generated with CoLLEGe, using the prompt "The word <nonce> is defined as". Each definition is generated using the single example sentence shown. None of the example sentences are provided in-context to the model.

GRE Verbal Reasoning problems with a single blank (i.e. not multi-part). No partial credit is given for selecting one of the two answers for a "top-2" question and chance accuracy is 8.1%. On a version of this task with the real word forms, not new tokens, a pretrained LLaMa-2 7B scores 75%, which serves as an upper bound on our potential performance.

To evaluate, we create a version of the sequence for each possible choice, and calculate the log probability of each such sequence. The highest log probability sequence is selected as the chosen answer. The final scores reflect the average accuracy over 10 trials of sampling different example sentences from GPT-4.

**Results:** Results are reported in Table 1, with additional results for "Prompting+..." reported in Table 8 in Appendix F.1. Token tuning does not seem to help much, and even sometimes hurts performance. Model performance also increases with more examples, showing effective utilization of multiple example sentences. By contrast, more examples can sometimes harm Prompting performance, likely due to the additional in-context examples distracting the LLaMA model. We ablate the use of negative examples, knowledge distillation, and both the $L_{\text{cos}}$ and $L_{\text{mse}}$ components the knowledge distillation loss. CoLLEGe outperforms each ablation except losing to one in two columns (D+2/3-shot). Notably, neither negative examples nor knowledge distillation is that effective on its own, but when combined boost performance significantly.

### 5.2 Definition Generation

To probe how well our model understands a new word, we prompt the LLM to generate a definition for the word given one or more example sentences, as shown in Figure 1.

**Datasets:** We evaluate on two different datasets to test definition generation: CoLLEGe-DefGen, which we create, and the Oxford dataset used in Giulianelli et al., 2023. To construct CoLLEGe-DefGen, we selected 954 words from WordNet (Miller, 1994). We then prompt GPT-4 (OpenAI, 2023) to generate an example sentence for the word using the prompt: *Give me a unique, descriptive sentence using the word "[WORD]" without defining it or making it obvious what the word means.* Without the latter half of the prompt, many generated examples rephrased the definition. Examples are generated at temperature = 0.8. Since both our model and the baselines continue to generate text, we select the first sentence generated as the definition for scoring.

**Results:** Generation examples are reported in Table 3. The generated embeddings often capture high- and low-level semantic details of the example sentences. Sometimes this is

| Model | CoLLEGe-DefGen | | Oxford | |
|---|---|---|---|---|
| | BERTScore F1 | ROUGE-L | BERTScore F1 | ROUGE-L |
| FLAN–Base–DefInstr | 84.0 | 12.1 | 83.4 | 15.4 |
| FLAN-Large–DefInstr | **85.0** | 13.2 | **83.6** | 16.6 |
| FLAN–XL–DefInstr | 83.7 | 11.6 | **83.6** | 14.0 |
| CoLLEGe | 83.5 | 16.5 | 83.1 | 15.4 |
| Prompting + CoLLEGe | 84.1 | **18.0** | **83.6** | **17.1** |

Table 4: Results for our comparison with Giulianelli et al. (2023)'s FLAN-T5 models on our CoLLEGe-DefGen dataset and the Oxford dataset. All evaluations are done 1-shot, with a single example sentence, and the target word is replaced with a new token.

fairly precise, for example the generated definition for *willies* is exactly correct and similarly with *copyist*. CoLLEGe is also able to identify *Wynnea americana* as a mushroom. Even when the definition is not quite right, it may capture general features of the concept correctly, and may reflect the limited information contained in the example sentence. The definition for *acupressure*, for example, is not exactly correct but a very good inference based on the example provided. Additional generated definitions, including those generated using more than one example sentence, are shown in Table 11 in Appendix G.1. We also show side-by-side comparisons with the baselines in Table 12 in Appendix G.2. Some failure cases are described in Appendix G.3.

In order to evaluate the quality of generated definitions, we compare a definition generated from our model to one generated from a baseline model as well as to a ground truth definition. Definitions are generated 1-, 2-, and 3-shot and results are averaged and reported in Table 2. For comparison between models, we simulate a head-to-head competition and compute the ELO score (Elo, 1978) of each model. Specifically, for each example in the task dataset, we choose a $k$-shot setting and sample a baseline at random. We then compare the definition generated by CoLLEGe with the one generated by the baseline. Based on the result—win, lose, or tie—we update the ELO score, starting with an initial score of 1000 for all models. To choose a winner in each "round", we use the Outlines package[4], we ask GPT-3.5 to select which definition is best for the word in question or if they are tied. The order of the choices (both generated definitions and "tie") are randomized. We compute ELO separately for Table 9. We also compare the generated definitions with ground truth definitions for each word by computing the BERTScore F1 (Zhang et al., 2020) and ROUGE-L (Lin, 2004). In all three quantitative evaluations, CoLLEGe outperforms the baselines. Qualitatively, only Prompting produces generated definitions that are somewhat competitive. Definitions generated by the other baselines are often incoherent, generating repetitive text or unrelated words and characters.

We next compare CoLLEGe with the series FLAN-T5 (Chung et al., 2024) models finetuned by Giulianelli et al. (2023) to generate definitions, which we denote FLAN-Base–DefInstr, FLAN-Large–DefInstr, and FLAN-XL–DefInstr. Each of these models is prompted with an example sentence followed by the question, *"What is the definition of [WORD]?"*. For a fair comparison, we replace the target word with a new token when evaluating the FLAN-T5 models and restrict CoLLEGe to one example sentence, since the finetuned FLAN-T5 models are trained to only use one example as well. Finally, since the FLAN-T5 models are prompted, we include results for Prompting + CoLLEGe as well. The results are presented in Table 4. When evaluated on our dataset, both CoLLEGe and Prompting + CoLLEGe achieve much higher ROUGE-L scores than all FLAN-T5 models. For BERTScore, Prompting + CoLLEGe outperforms all but FLAN-Large, although all scores are fairly close together. On the Oxford dataset, CoLLEGe and Prompting + CoLLEGe score comparably to the FLAN-T5 models, and Prompting + CoLLEGe slightly outperforms in terms of ROUGE-L. Notably, CoLLEGe achieves this zero-shot, without any additional training on the definition generation task, while the FLAN-T5 models are specialized for this purpose.

### 5.3 Twitter Slang

To emulate new word learning in a more natural setting, we construct a task based on identifying the correct definition for a slang term, using Tweets as example sentences.

---

[4]`https://github.com/outlines-dev/outlines`

| Model | 1-Shot | 2-Shot | 3-Shot | 4-Shot |
|---|---|---|---|---|
| TT-1 | $32.2 \pm 1.4$ | $32.1 \pm 2.6$ | $32.6 \pm 1.3$ | $34.5 \pm 0.4$ |
| TT-2 | $32.3 \pm 1.6$ | $32.8 \pm 3.9$ | $33.1 \pm 0.2$ | $32.5 \pm 3.0$ |
| Additive | $27.0 \pm 1.0$ | $28.3 \pm 1.9$ | $28.0 \pm 0.7$ | $29.0 \pm 1.0$ |
| HiCE | $34.0 \pm 1.1$ | $32.7 \pm 1.9$ | $31.3 \pm 2.3$ | $32.8 \pm 1.0$ |
| Prompting | $41.0 \pm 1.0$ | $47.0 \pm 2.1$ | $51.7 \pm 1.8$ | $53.8 \pm 1.4$ |
| CoLLEGe | $\mathbf{49.3} \pm 1.0$ | $\mathbf{53.2} \pm 1.6$ | $\mathbf{54.8} \pm 2.6$ | $\mathbf{60.0} \pm 0.7$ |

Table 5: Accuracy in percentage on the CoLLEGe-Slang benchmark dataset. CoLLEGe achieves higher accuracy on our Twitter Slang task than each of the baselines, including Prompting.

**Dataset:** To build CoLLEGe-Slang, we first hand-curate a set of 80 recent slang terms as well as their definitions. Alongside each term is a list of up to 8 high-quality example Tweets which use the term in an informative way. For this hand-curated set, example tweets are predominantly from 2022 and 2023. To supplement these, we then sample 120 additional slang terms from UrbanDictionary and the Online Slang Dictionary. We select example tweets from the Twitter archive, using the pipeline from Hu et al. (2022). Some examples from the hand-crafted set are shown in Table 13. More information about the filtering process, data sources, and curation details for this dataset can be found in Appendix D.

To evaluate the different models on this task, we select the true slang term, its example tweets, and its true definition. We then select 3 different incorrect slang terms and their examples. We score the log probability of the true definition conditioned on each set of examples. The highest probability is selected as the "choice". If it corresponds to the correct combination of definition and slang term, that is counted as a correct choice, otherwise not. We score the model based on its accuracy across the whole set of slang terms.

**Results:** Results are presented in Table 5. Without providing example tweets in-context, CoLLEGe outperforms each baseline. In fact, CoLLEGe is able to outperform prompting directly, showing that in novel contexts (such as Twitter slang), in-context examples may be more confusing than a concise embedding representation. Notably, when including example tweets in-context, the baselines hurt performance compared to a simple prompting baseline. Only using the CoLLEGe generated embeddings in this setting improve performance over prompting with randomly initialized new token embeddings. We also compare CoLLEGe to our baselines in the prompting setting in Table 10.

## 6 Conclusion

In this paper we present CoLLEGe, a few-shot learning framework for new concept acquisition for pretrained LLMs. We model our meta-learning approach on the original pretraining task by sampling few-shot learning episodes from language model pretraining datasets and using next-word prediction, as our primary meta-learning objective. CoLLEGe generated embeddings contain rich and task-general semantic information, generalize to multiple challenging tasks at test time zero-shot, without additional finetuning or training, and are particularly useful for more complex reasoning tasks.

## 7 Limitations and Future Work

While CoLLEGe achieves the best performance in all of the benchmarks, we summarize a few limitations in our current framework. First, the generated embeddings sometimes miss precise details in the examples, instead encoding higher level semantic information. This can be seen in some incorrect generated definitions, where general features of the unknown concept are correctly inferred even though specific details are missed. Second, we find the averaging mechanism cannot fully achieve parity with pretrained embeddings, even with more support sequences.

Our work points to a number of future research directions. In the short term, work needs to be done to investigate different data mixes for training CoLLEGe. More broadly, this research is a first step in an exciting direction for future research: online continual concept acquisition performed jointly with pretraining—incrementally identifying and compressing new concepts from an online stream of sequential experience.

## Acknowledgments

We would like to thank the Microsoft Accelerating Foundation Models Research program for providing cloud compute credits for running some parts of our LLM experiments. The compute was also supported by the NYU High Performance Computing resources, services, and staff expertise.

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

## A  Dataset Composition

Our dataset for training CoLLEGe consists of samples drawn from subsets of the Pile shown below in Table 6.

| Source | Num. Examples |
|---|---|
| Pile-CC | 79,606 |
| Books3 | 51,850 |
| BooksCorpus2 | 3,436 |

Table 6: The top Pile subsets represented in our dataset.

## B  Implementation Details

During training, we provide 1 support sequence for each query sequence and generalize to $K > 1$ during testing. We train our model for 28000 steps at batch size 32, with a learning rate of 1e-3, a linear learning rate schedule with warmup, and using the AdamW optimizer (Loshchilov & Hutter, 2019) with weight decay = 0.1 and default beta values. We experimented with different beta values during training, but found they had little effect. During training we clip gradients to a norm of 1.0. The final model checkpoint is selected based on its GRE score.

Using the default initialization for our Encoder produces input and output embeddings that are significantly larger in norm than those of the pretrained LLaMA model. During training with the default initialization, a lot of training time is spent reducing the norm. To address this inefficiency, we apply a Layer Normalization (Ba et al., 2016) layer before the input and output linear layers, and initialize those layers so that the expected norm is as close to the average input or output token embedding norm as possible.

## C  GRE Verbal Reasoning Examples

The GRE task consists of two different types of fill-in-the-blank questions. The first type asks you to select the best possible choice to complete the provided sentence, so it serves as a test of the top-1 prediction. The second type asks for which two words best complete the sentence. Often these words are similar, but distinct. It tests how the top-2 predictions of the LLM. We show an example for both types below in Table 7.

## D  Data Processing

We provide further details on cleaning and processing for some of our task datasets.

| Question | Answer Choices | Correct Answer(s) | Evaluation Type |
|---|---|---|---|
| Mary's former classmates were taken aback by her [BLANK] behavior at the reunion for, during her school years, she was frequently reprimanded for creating disturbances with her exuberant outbursts and playful antics. | a) gregarious
b) discourteous
c) obsequious
d) reticent
e) scurrilous | d) reticent | Choose the word for each blank that best fits the meaning of the sentence as a whole. |
| The firefighter, desperate to save the children on the second floor of the fiery house, rushed into their bedroom; his colleagues, more wary of the [BLANK] structure, remained outside. | a) stalwart
b) precarious
c) stout
d) irrefragable
e) tottering
f) fecund | b) precarious
e) tottering | Select the *two* answer choices that, when inserted into the sentence, fit the meaning of the sentence as a whole and yield complete sentences that are similar in meaning. |

Table 7: The two types of questions used in CoLLEGe-GRE.

**GRE:** The cleaning process for the GRE dataset involved normalizing the different formats for blanks (i.e. empty spaces, underlines, "(a)", etc.), removing artifacts from the conversion to text from PDF, and associating each question with its answer from the answer key.

**Twitter Slang:** We filter examples from this archive for flagged obscene words using Squeakily, but it is important to note that online slang is often obscene. This is especially true for the sources used to define the slang terms (UrbanDictionary in particular).

We used slang definitions from UrbanDictionary[5] Dictionary.com's Pop Culture[6] and Slang[7] sections, the recent American Dialect Society meeting[8], Bark [9], Wiktionary[10], and the Online Slang Dictionary[11] for our dataset.

# E  Baseline Implementations

**Hice:** To train HiCE, we follow the method outlined in the paper, and use the WikiText-103 dataset to train the model with a morphology network. We use hyperparameters from the authors' implementation.

**Word2Vec Projection:** For the Additive baseline as well as HiCE, the baseline outputs a Word2Vec embedding. To make this compatable with LLaMa, we train a linear layer on shared tokens between LLaMa and Word2Vec to map between the embedding spaces.

**Token Tuning:** For Token Tuning, we treat the example sentences as a "batch" and perform N=1,2 steps of gradient descent on only the input and output embeddings for the new token.

# F  Results for Prompting+

Since our **Prompting** baseline provides all the support sentences in the context window, allowing the LM to attend to the new token embeddings directly, we also show using CoLLEGe-generated embeddings improves performance over random initialization and our other baselines (denoted "Prompting+...").

---

[5]https://www.urbandictionary.com/

[6]https://www.dictionary.com/e/pop-culture/

[7]https://www.dictionary.com/e/slang/

[8]https://americandialect.org/nominations-for-words-of-the-year-2023/

[9]bark.us

[10]https://www.wiktionary.org/

[11]http://onlineslangdictionary.com/

### F.1 GRE Results for Prompting+

Table 8 shows results for the GRE task when prompting with examples in-context, using embeddings for the new token generated by the model or baseline following the "+".

| Method | Without Definition | | | | With Definition | | | |
|---|---|---|---|---|---|---|---|---|
| | 1-Shot | 2-Shot | 3-Shot | 4-Shot | D+1-Shot | D+2-Shot | D+3-Shot | D+4-Shot |
| Prompting | $13.9 \pm 4.3$ | $17.7 \pm 3.0$ | $19.8 \pm 4.2$ | $21.8 \pm 3.0$ | $21.6 \pm 5.7$ | $20.5 \pm 6.8$ | $19.3 \pm 4.8$ | $24.0 \pm 4.7$ |
| + CoLLEGe w/o KD / Neg. Ex. | $25.7 \pm 3.9$ | $31.1 \pm 3.9$ | $32.7 \pm 5.7$ | $32.1 \pm 6.1$ | $35.4 \pm 4.3$ | $31.8 \pm 3.8$ | $33.4 \pm 4.8$ | $\mathbf{33.4} \pm 5.8$ |
| + CoLLEGe w/o KD | $26.1 \pm 4.5$ | $29.6 \pm 3.4$ | $31.8 \pm 4.7$ | $29.1 \pm 4.0$ | $33.9 \pm 4.7$ | $28.9 \pm 4.1$ | $\mathbf{33.9} \pm 4.7$ | $35.7 \pm 2.3$ |
| + CoLLEGe w/o $L_{cos}$ | $25.9 \pm 5.0$ | $29.3 \pm 6.5$ | $28.9 \pm 3.7$ | $30.7 \pm 4.8$ | $33.4 \pm 6.6$ | $32.3 \pm 4.7$ | $32.5 \pm 2.9$ | $34.1 \pm 3.5$ |
| + CoLLEGe w/o $L_{mse}$ | $31.6 \pm 5.6$ | $31.8 \pm 5.8$ | $29.6 \pm 3.9$ | $27.5 \pm 5.4$ | $31.5 \pm 5.6$ | $31.8 \pm 5.8$ | $29.6 \pm 3.9$ | $31.8 \pm 3.8$ |
| + CoLLEGe w/o Neg. Ex. | $\mathbf{35.0} \pm 3.7$ | $\mathbf{33.2} \pm 2.3$ | $31.4 \pm 3.6$ | $25.9 \pm 10.3$ | $31.1 \pm 7.1$ | $31.1 \pm 4.6$ | $28.6 \pm 3.7$ | $28.4 \pm 4.9$ |
| + CoLLEGe | $34.3 \pm 4.3$ | $33.0 \pm 5.6$ | $\mathbf{34.8} \pm 3.4$ | $\mathbf{33.6} \pm 3.6$ | $\mathbf{37.5} \pm 5.6$ | $\mathbf{34.1} \pm 3.8$ | $30.5 \pm 4.6$ | $\mathbf{33.4} \pm 4.6$ |

Table 8: Accuracy in percentage on the GRE Verbal Reasoning task when prompting with examples in-context, using new token embeddings generated by each model or baseline. Results for prompting with randomly initialized embeddings are reproduced here for clarity.

All CoLLEGe models outperform the Prompting baseline, where new token embeddings are randomly initialized. Prompting by including the definition of each term alongside the example sentences improves performance for the baselines the most, but CoLLEGe significantly outperforms. Each "Prompting+CoLLEGe" model performs worse than the unprompted version, which we hypothesize is due to a lack of instruction tuning data in the training dataset.

### F.2 Definition Generation Results for Prompting+

We present results in Table 9 for our definition generation task with examples presented in-context, using our baselines or CoLLEGe to generate the new token embedding. ELO scores are calculated separately from those in Table 2 by sampling a random baseline challenger to the "Prompting+CoLLEGe" model.

| Model | BERTScore F1 | ELO | |
|---|---|---|---|
| Prompting | $0.825 \pm 0.028$ | 1002.09 | $\pm$ 22.87 |
| + TT-1 | $0.762 \pm 0.035$ | 959.88 | $\pm$ 17.22 |
| + TT-2 | $0.763 \pm 0.045$ | 962.10 | $\pm$ 12.65 |
| + HiCE | $0.740 \pm 0.098$ | 949.32 | $\pm$ 10.77 |
| + Additive | $0.735 \pm 0.098$ | 950.61 | $\pm$ 3.48 |
| + CoLLEGe | $\mathbf{0.858} \pm 0.029$ | **1176.00** | $\pm$ 12.69 |

Table 9: Results for the definition generation task, when prompting with examples in-context. We compare the model generated definitions with a reference definition generated by GPT-4 using BERTScore and simulate random challenges between CoLLEGe and each baseline and compute and ELO rating. Token Tuning results are for LR = 3e-4, as in the main paper.

Generated definitions improve with in-context examples, and we note that our model far outperforms the baselines. The only competitive baseline is prompting with randomly initialized embeddings.

### F.3 Twitter Results for Prompting+

When examples Tweets are provided in-context, our CoLLEGe model's accuracy on the slang identification task increases. For other baselines, aside from prompting with randomly initialized embeddings, however, performance either degrades or remains about the same. With the Word2Vec-based baselines, this may be due to the difficulty of mapping between Word2Vec embeddings and the LLaMA input and output embedding space. We present these results in Table 10.

## G Generated Definitions

### G.1 Additional CoLLEGe Definitions

We show additional definitions generated from CoLLEGe, including definitions generated with more than one example, in Table 11.

| Model | 1-Shot | 2-Shot | 3-Shot | 4-Shot |
|---|---|---|---|---|
| Prompting | 41.0 ± 1.0 | 47.0 ± 2.1 | 51.7 ± 1.8 | 53.8 ± 1.4 |
| + TT-1 | 30.5 ± 3.5 | 28.2 ± 2.9 | 26.0 ± 1.4 | 24.1 ± 1.2 |
| + TT-2 | 29.3 ± 3.0 | 28.3 ± 2.7 | 25.8 ± 3.1 | 24.0 ± 1.6 |
| + Additive | 30.7 ± 3.8 | 25.2 ± 0.4 | 24.5 ± 1.2 | 24.1 ± 1.7 |
| + HiCE | 25.0 ± 4.9 | 26.0 ± 1.1 | 27.8 ± 3.0 | 25.5 ± 0.8 |
| + CoLLEGe | **56.5** ± 2.0 | **60.5** ± 2.1 | **67.4** ± 1.0 | **69.8** ± 0.8 |

Table 10: Accuracy in percentage on the Twitter Slang task, where example Tweets are provided in-context.

| Example Sentence | CoLLEGe Definition | True Definition | Word/ Phrase |
|---|---|---|---|
| During the complex abdominal surgery, the surgeon carefully moved the <nonce> aside to gain better access to the patient's damaged organs. | a surgical procedure in which a portion of the intestine is brought through an opening in the abdominal wall. | a fold of peritoneum supporting the viscera | *omentum* |
| The yellow blooms of the <nonce> added a vibrant contrast to the green canvas of the wetlands. | "a plant of the genus Ficus, having a milky sap and large, often edible, fruit." | aromatic evergreen or deciduous dioecious shrubs or trees of eastern Asia and North America | *Lindera* |
| The prestigious <nonce>, clad in elaborate costumes, filled the auditorium with their mesmerizing harmonies and dramatic performances. After months of rigorous rehearsals, the <nonce> finally brought their magnum opus to life, filling the ornate theater with powerful harmonies that resonated with every member of the riveted audience. | a person who is skilled in the art of dancing. | a company that produces operas | *opera company* |
| Despite countless imitations flooding the market, only her grandmother's secret recipe for apple pie was the <nonce>. After tasting many alternatives, he finally found the <nonce> of artisanal cheeses in a quaint little shop in Paris. Despite all the replica paintings she had seen, it was breathtaking to stand before the <nonce> in the museum. | the most beautiful or perfect specimen of its kind. | informal usage attributing authenticity | *real Mc-Coy* |

Table 11: Additional definitions generated with CoLLEGe, using the prompt "The word <nonce> is defined as". Each definition is generated using the examples. None of the example sentences are provided in-context to the model.

## G.2 Qualitative Comparison of Generated Definitions

We present the definitions generated by CoLLEGe side-by-side with those generated by our baselines in Table 12.

| Word/Phrase | *horsecar* | *popishly* |
|---|---|---|
| **True Definition** | an early form of streetcar that was drawn by horses | like the Pope; in a popish manner |
| **CoLLEGe** | a "motorized vehicle with a cabin and a platform for passengers to stand on." | in a manner that is intended to attract attention or admiration |
| **HiCE** | a word" "the a word a" is a | a " a "a" a a word a word a a a |
| **Additive** | that the place where you are the place. | that the same. |
| **TT-1** | follows. | 'the opposite of the word " " in the dictionary |
| **TT-2** | follows. | the opposite of the word, |
| **Prompting** | in the heart of bustling 19th century Boston, he spent his mornings as a loyal conductor, navigating through | a flamboyant manner of acting or speaking. |

Table 12: Side-by-side comparison between CoLLEGe-generated definitions and definitions generated by the baselines.

In general, the baselines (without Prompting) are unable to generate usable definitions for the new concept.

### G.3 Failures of Generated Definitions

**Word2Vec Baselines:** Both Word2Vec baselines tended to produce embeddings that were unusable for generating definitions.

**CoLLEGe Failures:** When analyzing generated definitions, it is clear that there are some "default" definitions the model will generate when the embedding for the new token is not informative enough.

Some of these common "default" definitions, listed in order of frequency, are:

- a person who is not a member of a particular group or class
- a place of refuge or shelter
- a person who is a source of annoyance or irritation
- a noun

## H   Slang Examples

We highlight a few Twitter slang examples from the hand-curated subset of our task dataset in Table 13.

| Slang Term | Definition | Example Tweet |
|---|---|---|
| *rizz* | The ability to confidently approach people and talk to them, in a more romantic or flirty way. | Imagine having so little *rizz* that even the AI girlfriend rejects you. Just complete negative game. |
| *hits different* | When something is significantly better than usual or is way better under certain circumstances. | getting called pretty in person just *hits different*. people be making my day. |
| *gorpcore* | A fashion style that is similar to that of hiking/wilderness/utility tech wear. | An anecdote from my coverage of *Gorpcore* as a trend: This vintage seller put 4 Gore-Tex hats up for sale on his website at $135.... |

Table 13: Examples from the Twitter Slang task, showing the slang term, its definition, and an example tweet.

