# OpenReview forum: "CoLLEGe: Concept Embedding Generation for Large Language Models"
_colmweb.org/COLM/2024/Conference — COLM_

### Official Review · Reviewer_zTnf · 2024-05-04

**Rating:** 7
**Confidence:** 3
**Ethics Flag:** 1

**Summary:**

This paper presents a framework for a concept embedding generation system. The proposed method is a rather elegant approach to learning new concept embeddings, where new concepts are learned by masking them with a frozen MLM (roBERTa), encoding example sentences mentioning them, pooling their representation, and then train an autoregressive LLM (LLama2-7b) with an augmented vocabulary of +1 new concept. Different components play an important role, such as sampling negative examples or ensuring the masked representation is as close as possible to the true (unmasked) one.

The method I found to be really neat, and the presentation and general writeup very clear. The experiments are also reasonable (verbal reasoning, definition generation and correct definition identificatino from Twitter), and show that such models could be very useful for domain-specific or slang understanding.

My main criticism is that the evaluation, while sensible, seems to be quite limited and not be well positioned with existing lexical semantics benchmarks. It seems this would be important so that we can understand, beyond comparing with GPT, etc., how this fine-tuned Llama-2 model fares with older but well tested models.

**Questions To Authors:**

Just a typo: Sect 5.1 "On an version"

**Reasons To Accept:**

Very nice method, interesting insight leading to the negative examples loss and knowledge distillation. This leads to flexible representations that could be used in several downstream tasks.

Very well written paper, with nice introduction and plenty of additional details in the appendices.

**Reasons To Reject:**

The evaluation is a bit slim and ad-hoc. For example:

- GRE experiment seems fine, however I was't able to find any discussion about the role of definitions, and given that increasing the number of definitions for the considered terms boosts performance significantly, it would be good to understand a bit better why. It is not clear if they all come from the same source, mixed, or if more than 4 were tested.

- Definition generation. There is a body of work on definition generation (aka definition modeling), with standard datasets and metrics. I would have expected these works and baselines to be included (see below).

- Twitter slang experiment seems fine, although again, there are other works and datasets that could have been referenced. Beyond this, this looks very similar to the WiC-TSV task, so I'd encourage the authors to add an experiment here and compare with appropriate baselines.


Definition generation
[1] - Bevilacqua, M., Maru, M., & Navigli, R. (2020). Generationary or “how we went beyond word sense inventories and learned to gloss”. In Proceedings of the 2020 Conference on Empirical Methods in Natural Language Processing (EMNLP) (pp. 7207-7221).
[2] - Giulianelli, M., Luden, I., Fernández, R., & Kutuzov, A. (2023). Interpretable word sense representations via definition generation: The case of semantic change analysis. arXiv preprint arXiv:2305.11993.
[3] - Mickus, T., Van Deemter, K., Constant, M., & Paperno, D. (2022). Semeval-2022 Task 1: CODWOE--Comparing Dictionaries and Word Embeddings. arXiv preprint arXiv:2205.13858.

WiC-TSV
[3] - Breit, A., Revenko, A., Rezaee, K., Pilehvar, M. T., & Camacho-Collados, J. (2020). WiC-TSV: An evaluation benchmark for target sense verification of words in context. arXiv preprint arXiv:2004.15016.
Mapping words to definitions and mentions

Other helpful works mapping words, definitions and examples, maybe good for swapping with the original encoder
[4] - Yu, W., Zhu, C., Fang, Y., Yu, D., Wang, S., Xu, Y., ... & Jiang, M. (2021). Dict-bert: Enhancing language model pre-training with dictionary. arXiv preprint arXiv:2110.06490.
[5] - Chen, P., & Zhao, Z. (2022). A unified model for reverse dictionary and definition modelling. arXiv preprint arXiv:2205.04602.

---

> ### Author Rebuttal · Authors · 2024-05-31
>
> We thank the review for their helpful comments on our work and for pointing to prior work relevant to our evaluations.
>
> > The evaluation is a bit slim and ad-hoc.
>
> All of our evaluations are done zero-shot, without any additional finetuning. We built our tasks to test a wide array of settings, including complex verbal reasoning, definition modeling, and a more natural slang setting. CoLLEGe is trained on general purpose language modeling data to enable broad transfer.
>
> > GRE experiment seems fine, however I was't able to find any discussion about the role of definitions, and given that increasing the number of definitions for the considered terms boosts performance significantly [...].
>
> To clarify, we only present the model with one definition alongside 1–4 support sequences. For GRE, we treat each answer choice as a new concept represented with a new token.
>
> > Definition generation [...]
>
> Thank you for the pointers to this research. With the exception of Dict-BERT, we were unaware of these, so this is very helpful. We reached out to the authors of Generationary, as [suggested](https://sapienzanlp.github.io/generationary-web/), but have yet to hear a response. We were able to compare our work with that of Giulianelli et. al. (2023) by generating definitions 1-shot from their Oxford dataset and replacing the target word with a new token. The results can be found [here](https://postimg.cc/K114Vk91). CoLLEGe and Prompting + CoLLEGe score comparably to FLAN models trained for definition generation when the target word is replaced with a new token.
>
> We also evaluated the FLAN models on our task, with results [here](https://postimg.cc/Tp43r2Rf). CoLLEGe outperforms in terms of ROUGE-L and performs on-par or slightly better than FLAN when evaluated with BERT-F1. Importantly, the FLAN models are trained specifically to generate definitions given an example sentence. CoLLEGe, on the other hand, does not undergo any additional training or finetuning on definition modeling and is able to perform about the same or better.
>
> >Twitter slang experiment [...]
>
> Thank you for the suggestion. Our Twitter task is related to WiC but distinct. WiC is binary classification, but in Twitter we provide a choice between one correct and three incorrect (slang, definition) pairs. We are in the process of comparing these methods to ours on both WiC and on our Twitter task, and we plan to add these additional results during discussion.

---

> > ### Comment · Reviewer_zTnf · 2024-06-06
> >
> > Thank you for the answers and the additional experiments, the comparison with the Flan definition generation models is insightful.

---

> > > ### Author Response · Authors · 2024-06-07
> > >
> > > Thank you for reading our comments and increasing your score.

---

### Official Review · Reviewer_EA4C · 2024-05-12

**Rating:** 7
**Confidence:** 4
**Ethics Flag:** 1

**Summary:**

This paper introduces a framework called CoLLEGe for learning how to embed concepts from LLMs. The framework uses a meta-learning approach and makes use of text data from pretrained LM datasets. The proposed method efficiently samples support and query sequences by using saved sentences with new tokens. The knowledge distillation process then involves matching the generated embeddings with the ground truth embeddings and logits.

**Questions To Authors:**

- Could the authors explain how the concept embedding is taken into account in the LLM to produce the definition?
- Could the authors add more clarification about the sampling process?

**Reasons To Accept:**

- the paper is well-written and easy to follow
- the proposed method is novel, and the methodology is clear
- experiments on GRE verbal reasoning, definition generation, and slang identification show the efficiency of the proposed framework.

**Reasons To Reject:**

- Several works have been proposed recently on learning static vector representations of concepts such as MirrorBERT, Numberbach, etc. Perhaps authors should discuss these concept embeddings and use them instead of simply averaging mentions.

- more detail about the sampling and knowledge distillation process in section 3

---

> ### Author Rebuttal · Authors · 2024-05-31
>
> We thank the reviewer for their helpful suggestions and comments on our work.
>
> > Several works have been proposed recently on learning static vector representations of concepts such as MirrorBERT, Numberbach, etc. Perhaps authors should discuss these concept embeddings and use them instead of simply averaging mentions.
>
> This is an interesting idea, which we may explore in future work. MirrorBERT seems like a promising embedding method and worth trying. Numberbatch embeddings may be less directly applicable, but they may be useful alongside contextual embeddings.
>
> > more detail about the sampling and knowledge distillation process in section 3
>
> We first select words to be used as new concepts during training based on word frequency in the original training corpus, since we want words which do not appear too often, because they would be too easy to learn, or words that are too rare, because their representation would be too noisy. For each such word, we identify pairs of query and support sequences.
>
> For distillation, we retain the query sequence with the original word form (e.g. for “My ◾is complaining I never have any time to relax”, the original would be “My **beige flag** is complaining I never have any time to relax”). The intuition is that CoLLEGe should represent “is complaining I never have any time to relax” in the query sequence similarly to how LLaMA would represent it in the original sequence. This encourages CoLLEGe to generate embeddings such that the output embeddings and the logits for that subsequence are close (in cosine distance and mean-squared error, respectively) to those of the LLaMA model for that same subsequence originally
>
> > Could the authors explain how the concept embedding is taken into account in the LLM to produce the definition?
>
> We sample from the LLM conditioned on the prompt "The word '◾` is defined as'" in order to generate a definition for the new concept. CoLLEGe learns to generate an embedding for the new concept token ◾, and that embedding is projected to the input and output embedding space of the LLaMA model. The LLaMA model then makes use of those embeddings when applying the embedding the prompt (and subsequent sampled tokens) and when it computes the output logits.
>
> > Could the authors add more clarification about the sampling process?
>
> We believe you are referring to the sampling process for the definition generation task. All of our definitions are generated using greedy decoding at temperature = 1.0.

---

> > ### Comment · Reviewer_EA4C · 2024-06-04
> >
> > Thank you for the answers.

---

### Official Review · Reviewer_6Rmw · 2024-05-12

**Rating:** 5
**Confidence:** 3
**Ethics Flag:** 1

**Summary:**

The paper proposed a few-shot method to enable a language model to learn new concepts on the fly.

**Questions To Authors:**

No other questions

**Reasons To Accept:**

The problem is very interesting and it is one of the critical problems that should be solved for language models.

**Reasons To Reject:**

I found the problem studied in the paper artificial. I have a few concerns:

New concepts come constantly and naturally and should be identified and learned naturally by the language model itself in the particular contexts of the concepts. Manually identifying such concepts is not practical. How do you know that the language model does not have a concept already in the system? After identifying such concepts, positive and negative examples have to be prepared in sophisticated ways in order to learn the concepts. Training even involves meta-learning, which makes the task even more difficult.

Experiments are conducted but again artificial.

The paper mainly described what the authors have done with gory details, but provided little principle.

There are two closely related areas of research that the paper has missed. The first area is model editing. A large body of literature on the topic is trying to do what you are doing. Another area is continual learning, which tries to ensure little negative interference to the language model when learning new knowledge or concepts. The following papers can give you some ideas about the existing work in these two areas.

Mitchell et al., Fast Model Editing at Scale, ICLR-2022

Wang et al. Knowledge Editing for Large Language Models: A Survey. arXiv:2310.16218 [cs.CL]

Ke et al. Continual Pre-training of Language Models. ICLR-2023.

---

> ### Author Rebuttal · Authors · 2024-05-31
>
> We thank the reviewer for their helpful comments about our work.
> > New concepts come constantly and naturally and should be identified and learned naturally by the language model itself in the particular contexts of the concepts.
>
> This is an interesting direction for future research and we believe CoLLEGe is an important first step in addressing it. In order to eventually identify and learn new concepts constantly, we first develop a method which can learn identifiable new concepts on the fly.
>
> > After identifying such concepts, positive and negative examples have to be prepared in sophisticated ways in order to learn the concepts.
>
> Our goal is to present a method which learns a new concept given only a few examples. To train our model, select words to be used as new concepts and train our model to predict query sequences using them with a few support sequences. The new concept is represented by a new token, guaranteeing that the language model has not encountered it before. Negative examples along and distillation are used to improve the generated embeddings.
>
> > Training even involves meta-learning, which makes the task even more difficult.
>
> We choose a meta-learning objective that blends into the pretraining objective, which is not much more difficult than standard autoregressive training and allows CoLLEGe to generate task-general embeddings with rich semantic knowledge.
>
> > How do you know that the language model does not have a concept already in the system?
> > Experiments are conducted but again artificial.
>
> In our Twitter evaluation, we hand-select very recent slang terms. Some incorrect answers to questions about slang terms generated by ChatGPT-3.5 and LLaMA-2 can be seen [here](https://postimg.cc/D8K8Bmhd). Additionally, our Twitter and GRE evaluations contain multi-word phrases as single concepts, which LLaMA-2 model has not seen. Finally, if LLaMA could simply make use of its pretraining knowledge, then the baselines would not perform so poorly.
>
> > The first area is model editing.
> > Another area is continual learning, [...]
>
> Thank you for these pointers. We intend to cite them as relevant background, but do not believe they are directly relevant. Applying model editing to the new concept learning setting would mean manually editing the LM for each new concept, which seems more artificial and sidesteps our research problem. Also, our problem is not continually adapting to new domains or tasks, like in Ke et. al.

---

> > ### Comment · Reviewer_6Rmw · 2024-06-05
> > **Thanks for the rebuttal**
> >
> > I don't agree with your reply to my final point. I think they are very relevant. Model editing is not done manually. Some model editing methods also use positive and negative examples. Your method has to deal with forgetting as in continual learning or model editing.
> >
> > Thanks for responding to my comments. Since you have addressed some of my concerns, I increased my score to 5.

---

> ### Author Response · Authors · 2024-06-07
>
> Thank you for your response and for increasing your score, as well as for suggesting we compare with Knowledge Editing methods. To address your concerns, we evaluated some of these methods on our GRE task. Here, we use the definition of the new concept for editing and also present support sequences in-context. IKE [1] uses in-context learning for knowledge editing, whereas ROME [2] and MEND [3] update the model weights. To train MEND, we use the zsRE dataset and we use the implementations provided in EasyEdit [4]. In the first table below, we provide only support sequences in-context and in the second we also provide definitions.
>
> |                  |       1 shot       |      2 shots       |      3 shots       |      4 shots       |
> | :--------------: | :----------------: | :----------------: | :----------------: | :----------------: |
> |     CoLLEGe      | **35.0 ± 5.6** | **40.5 ± 4.3** | **42.7 ± 3.9** | **44.5 ± 3.9** |
> |    Prompting     |   13.9 ± 4.3    |   17.7 ± 3.0    |   19.8 ± 4.2    |   21.8 ± 3.0    |
> | Prompting + MEND |   13.4 ± 5.0   |   10.1 ± 2.9   |   14.7 ± 5.4   |   19.4 ± 1.9   |
> | Prompting + IKE  |   18.1 ± 2.7   |   22.3 ± 3.1   |   23.2 ± 3.2   |   27.9 ± 5.9   |
> | Prompting + ROME |   14.9 ± 5.2   |   16.2 ± 3.2   |   16.7 ± 5.3   |   22.3 ± 1.8   |
>
>
> |                  |   Def. + 1 shot    |   Def. + 2 shots   |   Def. + 3 shots   |   Def. + 4 shots   |
> | :--------------: | :----------------: | :----------------: | :----------------: | :----------------: |
> |     CoLLEGe      | **42.5 ± 4.3** | **46.8 ± 3.7** | **45.2 ± 3.2** | **49.3 ± 1.5** |
> |    Prompting     |   21.6 ± 5.7    |   20.5 ± 6.8    |   19.3 ± 4.8    |   24.0 ± 4.7    |
> | Prompting + MEND |   23.3 ± 5.0   |   10.8 ± 4.0   |   16.3 ± 1.9   |   20.2 ± 2.2   |
> | Prompting + IKE  |   20.0 ± 6.2   |   17.2 ± 5.2   |   17.6 ± 2.3   |   23.7± 5.8    |
> | Prompting + ROME |   21.9 ± 6.7   |   18.6 ± 5.1   |   14.4 ± 4.7   |   23.2 ± 3.2   |
>
> We find that knowledge editing methods score better than our Prompting baseline in some cases, when the definition is not provided in-context, but around the same or slightly worse when it is provided. In either case, CoLLEGe achieves much higher accuracy than all KE methods.
>
> We also tried using the support sequences as well as the definition during editing. Unfortunately, the time it took to evaluate ROME in this setting was much longer, and we were only able to complete 2 evaluation runs (as opposed to 5 above). Nonetheless, we found that performance was sometimes a bit worse than if we just use the definition for editing. As above, in the first table we provide only support sequences in-context and in the second we also provide definitions. In both cases, definitions and support sequences are used for editing.
>
>
> |      Method      |       1 shot        |       2 shots       |       3 shots       |       4 shots       |
> | :--------------: | :-----------------: | :-----------------: | :-----------------: | :-----------------: |
> |     CoLLEGe      | **35.0 ± 5.6** | **40.5 ± 4.3** | **42.7 ± 3.9** | **44.5 ± 3.9** |
> |    Prompting     |   13.9 ± 4.3    |   17.7 ± 3.0    |   19.8 ± 4.2    |   21.8 ± 3.0    |
> | Prompting + ROME |   10.4 ± 1.1    |   14.0 ± 2.3    |   12.8 ± 1.2    |   20.9 ± 0.0    |
>
> |      Method      |   Def. + 1 shot    |   Def. + 2 shots   |   Def. + 3 shots   |   Def. + 4 shots   |
> | :--------------: | :----------------: | :----------------: | :----------------: | :----------------: |
> |     CoLLEGe      | **42.5 ± 4.3** | **46.8 ± 3.7** | **45.2 ± 3.2** | **49.3 ± 1.5** |
> |    Prompting     |   21.6 ± 5.7   |   20.5 ± 6.8   |   19.3 ± 4.8   |   24.0 ± 4.7   |
> | Prompting + ROME |   26.7 ± 1.2   |   16.3 ± 4.7   |   11.6 ± 2.3   |   22.1 ± 3.5   |
>
> We find that ROME performs worse than Prompting in most cases, and each achieves lower accuracy than CoLLEGe.
>
> [1] Zheng, Ce, et al. "Can We Edit Factual Knowledge by In-Context Learning?." *Proceedings of the 2023 Conference on Empirical Methods in Natural Language Processing*. 2023.
>
> [2] Meng, Kevin, et al. "Locating and editing factual associations in GPT." *Advances in Neural Information Processing Systems* 35 (2022): 17359-17372.
>
> [3] Mitchell, Eric, et al. "Fast model editing at scale." *arXiv preprint arXiv:2110.11309* (2021).
>
> [4] Wang, Peng, et al. "Easyedit: An easy-to-use knowledge editing framework for large language models." *arXiv preprint arXiv:2308.07269* (2023).

---

> ### Author Response · Authors · 2024-06-07
>
> During the evaluation, we noticed that KE methods took much longer to evaluate than CoLLEGe. To quantify this, we computed the average amount of time each method took on each GRE evaluation example, which we present below. CoLLEGe and Prompting + CoLLEGe are 7x and 3x, respectively, faster than the fastest KE method, MEND. CoLLEGe is also 50x faster than the slowest method, IKE. One caveat is that we are using the implementation of these methods from EasyEdit and did not separately optimize their efficiency.
>
> |       Method        | Avg. Time Elapsed Per Example (s) &#8595; |
> | :-----------------: | :---------------------------------------: |
> |       CoLLEGe       |                   0.51                    |
> | Prompting + CoLLEGe |                   0.96                    |
> |  Prompting + MEND   |                   3.59                    |
> |  Prompting + ROMEO  |                   14.0                    |
> |   Prompting + IKE   |                   25.8                    |

---

### Decision · Program_Chairs · 2024-07-10

**Decision:**

Accept

**Comment:**

This paper proposes a new method for large language models to learn new concepts on the fly to overcome current methods such as prompt-based in-context learning and few-shot fine-tuning. For achieving this, the authors design a meta-learning approach for flexible embeddings for new coming concepts.

Among three reviewers, two reviewers voted to acceptance score (7) and one reviewer gave a rejection (4) initially.
All reviewers agree that interesting setup, the importance of the task that this paper addresses, novel methods, and good writing.
The main concerns were less natural setup and lack of description on related research area.

During the rebuttal period, the authors tried to address the concerns and it seems that some of them were alleviated.

AC carefully read the paper, the review comments, and the authors' feedback.

Considering the discussion comments as well, AC agree the contribution of this paper, and thus recommends accepting this paper.

AC asks the authors reflect valuable comments of the reviewers in the final camera-ready version.